# High-Efficiency Water Recovery from Urine by Vacuum Membrane Distillation for Space Applications: Water Quality Improvement and Operation Stability

**DOI:** 10.3390/membranes12060629

**Published:** 2022-06-17

**Authors:** Fei Wang, Junfeng Liu, Da Li, Zheng Liu, Jie Zhang, Ping Ding, Guochang Liu, Yujie Feng

**Affiliations:** 1School of Environment, Harbin Institute of Technology, No. 73 Huanghe Road, Nangang District, Harbin 150090, China; pheinix@126.com (F.W.); richard@hit.edu.cn (J.L.); dali@hit.edu.cn (D.L.); 6282031@163.com (J.Z.); 2National Key Laboratory of Human Factors Engineering, China Astronaut Research and Training Center, Beijing 100094, China; pingding2022@126.com; 3The Institute of Seawater Desalination and Multipurpose Utilization, MNR (Tianjin), Tianjin 300192, China; liuzheng@isdmu.com.cn (Z.L.); liuguochang_001@163.com (G.L.)

**Keywords:** water reclamation, stabilized urine, vacuum membrane distillation, membrane wetting, membrane fouling

## Abstract

Water recovery by membrane distillation (MD) is an attractive alternative to existing urine treatment systems because it could improve the water recovery rate and reliability in space missions. However, there are few studies of urine MD, particularly on the removal of the remaining contaminants from distillate water and the assessment of its long-term performance. In this study, the influences of various operation parameters on distillate water quality and operation stability were investigated in batch mode. The low pH of feedstock reduced the conductivity and total ammonium nitrogen (TAN) in distillate water because the low pH promoted the ionization of ammonia to ammonium ions. However, the low pH also facilitated the formation of free chlorine hydride, which resulted in the minor deterioration of the conductivity in the distillate due to the increasing volatility of chlorine hydride in the feedstock. Thirty batches of vacuum membrane distillation (VMD) experiments demonstrated that the permeate flux and the distillate water quality slightly decreased due to the small range of membrane wetting but still maintained an over 94.2% and 95.8% removal efficiency of the total organic carbon (TOC) and TAN, and the conductivity was <125 μs cm^−1^ in the distillate water after 30 test batches. VMD is a feasible option for urine treatment in space missions.

## 1. Introduction

Water recovery from urine is inevitable in extreme environments such as long-term, manned space exploration [1,2,3]. Although some technologies have been utilized in the International Space Station (ISS), novel approaches are still desired due to the suboptimal water recovery rate and the poor reliability of the established technologies [4]. Approximately 1500 kg of urine was excreted by each crew on a 30-month-long mission [4], and it was difficult to recover potable water by conventional treatment due to the extremely complex components and the physical–chemical properties of urine and the need for water recovery efficiency [5]. At present, the urine treatment system in the ISS uses vapor compression distillation (VCD) technology, although water recovery of <70% results in excessive unrecyclable brine and resupply for space missions [4]. Moreover, the VCD system is not reliable (e.g., clogging and leakage) due to its complicated mechanical structure [4]. Accordingly, many methods for the contaminant removal and water reclamation from urine have been studied, including advanced oxidation processes [6,7], biological and enzymatic methods [8,9], and membrane technology [10,11,12,13].

By contrast, the membrane-based approaches seem more suitable for water reclamation from urine because the technologies perform under controllable conditions with a membrane module (simple structure and high reliability) [4]. Among them, reverse osmosis (RO) was first studied for water recovery from urine. However, the RO module was unusable after being used once due to the excessive organic foulants in the urine [14]. Studies of urine treatment by forward osmosis (FO) indicated that a water recovery rate of <15% and draw solutions contaminated by permeable organics were the main obstacles for the applications [12,15]. Recently, membrane distillation (MD)—which uses the diffusive and convective transfer of vapor across the micropores on a hydrophobic membrane, rather than pressure driving—was considered as a suitable option for treating wastewater of high salinity and organic load [16]. Unlike RO and FO, in which the osmotic pressure restricts the water recovery rate, MD can achieve a high water recovery rate by vapor permeation. In addition, the membrane fouling of MD is milder than the other membrane separation technologies that depend on pressure driving [17]. A polyvinylidene fluoride/polytetrafluoroethylene composite membrane was developed for water recovery from hydrolyzed human urine by direct contact membrane distillation (DCMD), based on which the rejection of ammonia and the TOC reached 95% and 98%, respectively [18]. However, Xu et al. studied water reclamation and nutrient concentration from hydrolyzed urine via DCMD and demonstrated the permeation of ammonia into the distillate—resulting in the contamination of the produced water [19]. Zhao et al. performed fundamental studies of water reclamation from urine by VMD, proposed the concept of critical fouling operating conditions, and presented methods for relieving the fouling during urine VMD [20].

At present, there is limited knowledge on identifying and removing the remaining contaminants in the distillate water and the long-term performance of urine MD. Although additional equipment and power consumption are required, VMD exhibits higher flux and thermal efficiency than the other operation modes, including DCMD and air gap membrane distillation. Furthermore, the desirable vacuum source and solar power [20] provide assistance to VMD in space applications. In this study, VMD was selected to systematically assess the water recovery from urine. First, the influences of the major operation parameters on urine VMD were evaluated in order to improve the distillate water quality and mitigate the fouling of the membrane. Second, the operation stability was studied by multi-batch VMD experiments in optimized conditions. Third, the phenomena and mechanisms of the performance decline were discussed. This study provides meaningful suggestions for the efficient reclaiming of water from stabilized urine over 30 d of use. 

## 2. Materials and Methods

### 2.1. Chemicals

All chemicals were purchased from Sigma–Aldrich Trading Co., Ltd. (Shanghai, China) and Sino-Pharm Chemical Reagent Co., Ltd. (Shanghai, China)

### 2.2. Urine Preparation

Given that the composition and the content of urine differ between individuals, an ersatz urine with fixed components was prepared in this study to eliminate the influence of feed differences on the experimental results. Appendix A summarizes the composition and concentration of the ersatz urine based on previous studies [13,21]. To investigate the effects of the volatile components on the distillate quality during urine VMD, the phenol exhibiting a greater concentration than the other volatile constituents was added as a representative in the ersatz urine [21]. The concentrations of total salt and TOC were 14,630 and 5060 mg L^−1^, respectively. The initial conductivity was 23.5 ± 1 ms cm^−1^, and the pH was adjusted to 6.0–7.0 by using 3 mol L^−1^ hydrochloric acid.

Urine stabilization is necessary when considering treatment of human urine; without stabilization, urea hydrolysis occurs during collection and storage. The purpose of urine stabilization included inhibiting the organic matter degradation and microbial proliferation, as well as preventing an increase in the pH [14]. The urine stabilization in use aboard the ISS is based upon adding chemicals, including hexavalent chromium as a bacteriostat and sulfuric acid as an acidifier [22]. In this study, the same stabilization method was performed; the concentration of hexavalent chromium was set at 5 mmol L^−1^. In addition, in accordance with the treatment process of urine on the ISS, ca. 17% flushing water was added to the urine prior to water recovery [22].

### 2.3. Membrane Fabrication and Characteristics

In MD, the hydrophobic porous membrane is the key component for sufficient vapor permeation and the complete rejection of nonvolatile solutes in the feed [23]. To date, no specific membrane has been provided for MD. In most cases, the membranes used for MD experiments have been fabricated with various polymers, including polypropylene (PP), polytetrafluoroethylene, and polyvinylidene fluoride [24]. Although PP exhibits comparatively low hydrophobicity, previous investigations have indicated that PP exhibits the best performance in MD due to its physical–chemical stability and easy processing [25,26,27,28]. Polypropylene (PP) is a semicrystalline polymer with a surface energy reaching 30.0 × 10^3^ N/m, which provides enough hydrophobicity for the microporous membrane applied to the MD process [29,30,31]. Considering sufficient handling capacity as well as the limits on the dimensions and mass for space applications, a homogeneous PP hollow fiber membrane with a high surface area to volume ratio was fabricated by melt-spinning and stretching in this study. The preparation of the PP hollow fiber membrane was divided into three steps. Firstly, the precursor hollow fibers were fabricated using a granular material of PP (Sinopec, Lanzhou, China), which was melted and spun from a hollow spinneret though an inner hole. Secondly, the cooled precursor fibers with semicrystalline microstructure were annealed at 100–110 °C for 30–60 min to improve the perfection of the crystalline structure. Thirdly, the three-dimensional meshy micropores were fabricated with parallel lamellar crystal separation by hot–cold stretch. Finally, the resulting hollow fiber membranes with micropores were re-annealed at 115–135 °C for 30–60 min. 

The liquid entry pressure of water (LEP_w_) was the dominant index for preventing feed leakage through the membrane pores at the operation pressure difference between the feed side and the permeate side [29]. In accordance with the Laplace equation (Appendix A), the LEP_w_ depends on the tortuosity factor of the pores, the contact angle between the solution and the membrane surface, the largest pore size, etc. Considering that the maximum pressure difference was 101.3 kPa when the permeated side was pumped to vacuum (0 kPa, absolute pressure), the LEP_w_ was designed at 200 kPa to provide a sufficient pressure margin. In addition, the vapor flux through the micropores depends on the mean pore size and porosity, which should be suitable for sufficient permeate flux. The aforementioned properties of the membrane were regulated by the fabrication parameters—including the pretreatment and stretch temperature as well as the stretch ratio and rate [32,33].

The morphology (Appendix A) and the composition of the membrane were analyzed with a field emission scanning electron microscope equipped with an energy-dispersive X-ray spectrometer (SEM–EDS, S4800, Hitachi, Japan). The membrane samples were sputter-coated with gold before analysis. The functional groups in the membrane samples were characterized by Fourier-transform infrared (FTIR) spectroscopy (Nicolet is 10, Thermo Fisher, Waltham, MA, USA) with a mercury–cadmium–telluride detector and a KBr beam splitter. The hydrophobic properties were characterized via contact angle measuring using 1-μL droplets (JC2000C5, Zhongchen, Shanghai, China). The membrane porosity and pore size were measured by gas–liquid transfer porosimetry (POROLUX 1000, Antwerp, Belgium) (Appendix A). The LEP_w_ was measured by gradually increasing the pressure on the feed side until the beginning of the liquid permeation [34].

### 2.4. Performance Evaluation of Deactivating Microorganisms

The effect of the bacteriostat on the deactivating microorganisms was assessed by the microbial proliferation level in stabilized urine (plate counts). The samples were collected from stabilized and non-stabilized ersatz urine after 7 d of storage in ambient conditions and were serially diluted and plated on aerobic count plates (Petrifilm-6406, 3M, St. Paul, MN, USA). The colony-forming units (CFU) per milliliter were determined by visual counts after the plates were incubated at 35 °C for 48 h.

### 2.5. VMD Experiments and Assessment Methods

VMD was performed with two similar tubular membrane modules (M1 and M2), which were assembled with different membrane numbers. The M1 module contained 80 membranes assembled in a 0.25 m-long tubular housing and was used for the compositional optimization of urine VMD. The M2 module contained 800 membranes with the same specification and was used for investigating the influence of the water recovery rate on the urine VMD and the operation stability. The effective membrane areas of the M1 and M2 modules were 0.031 and 0.31 m^2^, respectively.

The experiments were performed with the experimental setup shown in Figure 1. The installation consisted of a thermostatic cycle (feed circulate) and a vacuum cycle (distillate flow), which were connected by a membrane module. All of the components and pipes in the feed circulate were covered with polyurethane foam as a thermal insulation layer. The membrane module was assembled in the vertical position in the VMD installation. The feed streams flowed upward from the bottom to the upper part of the VMD module. The feed flow rate was 60 L h^−1^; thus, the velocities for the M1 and M2 module were 1.10 and 0.11 m s^−1^, respectively. During the experiments, the feed was supplied inside the membrane capillaries (tube side) with a magnetic pump. Meanwhile, the distillate flowed on the shell side of the VMD module. The feed temperature was feedback-controlled with thermometers at a ±1 °C controlled precision. The permeate side pressure (shell side) was controlled with a vacuum pump and regulated with a vacuum transducer at a ±0.5 kPa controlled precision. The permeate vapor was condensed with a condensing coil connected in a cryostat at 5 °C. The condensed permeate stream was collected with a distillate tank placed on an online electronic balance.

The VMD was operated in batch mode; 12 L of stabilized urine per batch was used to simulate the total quantity of urine of 6 crews/d [4]. The water recovery rate of the multi-batch VMD experiments was fixed at ca. 80% to relieve the fouling deposited on the feed surface. Each batch was operated until the water recovery rate reached the intended target (ca. 80%) during the day. During the night, the setup was inoperative, and the feed solution was drained out of the modules. In addition, to eliminate the influence of the feed concentration on the permeate flux and the distillate water quality, the water recovery rate was set to be <30%, while the influence of the operating conditions (e.g., feed temperature, pH, and permeate side pressure) on the VMD was investigated. The tests of the permeate flux were performed at least three times, and the average was used for the discussion.

The trans-membrane distillate flux of the VMD is described by Equation (1) [12].
(1)$$J_{W}=\frac{V_{t\left(n+1\right)}-V_{t\left(n\right)}}{At}$$
where *J*_W_ (L m^−2^ h^−1^) is the permeate flux of the VMD, *V_t_*_(n)_ (L) and *V_t_*_(n+1)_ (L) are the volume of the permeate flux at the nth and (n + 1)th times of measurement, respectively, *t* (h) is the time interval of every measurement, and *A* (m^2^) is the effective membrane area.

The water recovery rate *W*_R_ (%) of the VMD is described by Equation (2) [13].
(2)$$W_{R}=\frac{V_{f,0}-V_{p,f}}{V_{p,t}}\times 100\%$$
where *V_f_*_,0_ (L) is the initial volume of the feed and *V_p_*_,*t*_ (L) is the permeate flux volume.

The contaminant removal rate is expressed as Equation (3) [12].
(3)$$R_{\mathrm{TOC}}=\frac{C_{\mathrm{TOC},f,0}V_{f,0}-C_{\mathrm{TOC},p,t}V_{p,t}}{C_{\mathrm{TOC},f,0}V_{f,0}}\times 100\%$$
where *C*_TOC,*f,*0_ (mg L^−1^) is the initial concentration of the TOC in the feed, *V_f,_*_0_ (L) is the initial volume of the feed, *C*_TOC,*p,t*_ (mg L^−1^) is the TOC concentration in the distillate water of the VMD, and *V_p,t_* (L) is the permeate volume. The same calculated method was used for the removal rate of TAN.

### 2.6. Analytical Methods

TOC was measured with a Sievers InnovOx ES TOC analyzer (Suez, Trevos, PA, USA). The TAN (including NH_4_^+^ and NH_3_) was measured colorimetrically by Nessler’s method at a wavelength of 420 nm. The cations (Ca^2+^, Na^+^, K^+^, and Mg^2+^) and anions (Cl^−^, NO_3_^−^, NO_2_^−^, and SO_4_^2−^) were measured by ion chromatography (Aquion, Dionex, Sunnyvale, CA, USA) with CS 12A and AS 14 columns, respectively. The Cr element was detected by inductively coupled plasma-optical emission spectroscopy (Optima 2100 DV, Perkin Elmer, Waltham, MA, USA). Phenol was measured by high-performance liquid chromatography (Ultimate 3000, Thermo Fisher, Waltham, MA, USA) with an ultraviolet detector. Methanol (30 *v*/*v*%) was used as the liquid phase, and the detection wavelength was 270 nm. Urea was detected by the diacetyl monoxime reagent colorimetric method. The electric conductivity and pH of the solutions were analyzed at 25 °C with a conductivity meter (model 3200/3252 probe, Yellow Springs, OH, USA) and a pH meter (model 7310/SenTix-41 probe, WTW, Weilheim, Bavaria, Germany). All tests were carried out at least twice. 

## 3. Results

### 3.1. Stabilized Urine

Table 1 shows the major components and their concentrations in fresh and stabilized ersatz urine. Compared with the fresh ersatz urine, the concentrations of TOC, urea, and TAN in the stabilized ersatz urine did not appreciably change, which demonstrates that the major nutrients were stable during storage. In accordance with the plate counts, the bacterial colonies in the non-stabilized urine reached 2.2 × 10^5^ CFU mL^−1^ after 7 d of storage, which demonstrates that microbes readily proliferated in urine when in the presence of abundant nutrients. However, no bacterial colonies were found in the stabilized urine (Appendix A). The low pH and increasing SO_4_^2−^ concentrations in the stabilized urine indicate the presence of sulfuric acid. The stabilized urine also exhibited higher conductivity than the non-stabilized urine. Moreover, a comparative decrease in the concentration of Ca^2+^ and total phosphorus (TP) was observed in the stabilized urine, which resulted from precipitation in accordance with the added sulfuric acid [4].

### 3.2. Influence of Temperature and Permeate Side Pressure on the Permeate Flux

The permeate flux increased with the inlet temperature and exhibited a negative relationship with the permeate side pressure (Figure 2). The test results demonstrate good agreement with the theoretical analysis in the Appendix A. In accordance with the theoretical analysis of water recovery by VMD, Knudsen diffusion dominated the mass transfer of vapor over a pore size range of 282 ± 19 nm. The mean pore size (Appendix A) measured by gas–liquid transfer porosimetry was used in the analysis and calculation because it was closely related with the vapor permeability of the porous membrane. When the VMD is operated at the same temperature, the permeate flux can be expressed by the following equation.
(4)$$N_{Kn}{=a\;(P}_{vapor,\;T_{m}}-{\;P}_{vacuum})$$
where *a* = $\left(8/3\right)\left[\left(\varepsilon r\right)/\left(\tau \delta \right)\right]\sqrt{1/2\pi RT_{m}M}$ is the slope of the *N_Kn_* − *P_vacuum_* curves.

The maximum flux reached 28.3 L m^−2^ h^−1^ at 7.3 kPa and 69 °C, whereas it obviously decreased with the increasing permeate side pressure and decreasing feed temperature. When the pressure of the permeate side increased to 19.3 kPa at 57 °C, no distillate was collected because the downstream pressure was higher than that at the evaporation surface, and essentially, no water vapor permeated through the membrane pores. The decomposition of the urea increased with the increasing temperature [12]. Therefore, the feed average temperature was determined to be 55 °C to 70 °C to achieve a satisfactory flux and yet inhibit the degradation of the urea. 

### 3.3. Influence of Temperature and Permeate Side Pressure on the Distillate Water Quality

The increasing temperature of the feedstock resulted in a slight deterioration of the TOC in the distillate. However, there was not an obvious influence of the permeated side pressure on the distillate water quality. The TAN was maintained at a relatively low level, and the removal efficiency was >99.5% at the initial pH of 1.6 over the tested range of permeate side pressures (Figure 3a) and temperatures (Figure 3b). This result is attributable to the fact that most of the ammonia was in the form of ammonium ions when the pH was <2 (Appendix A).

The TOC concentration in the distillate water was dominated by the transfer of phenol across the porous membrane as volatile molecules (Section 3.4). The concentration of the phenol in the distillate was determined by the separation factor *β* [35].
(5)$$\beta =\frac{x_{o,ph}/x_{o,w}}{x_{i,ph}/x_{i,w}}$$
where *x_o,ph_* and *x_o,w_* are the mole fractions of phenol and water in the distillate water, respectively, and *x_i,ph_* and *x_i,w_* are the fractions of phenol and water in the feed, respectively. When β approached 1, suggesting a similar proportion of phenol and water in the permeate flux and feedstock, the phenol could not be concentrated in either side. Otherwise, the phenol was concentrated in the permeate flux (*β* > 1) or feed (*β* < 1). The concentration of the remaining TOC in the distillate was higher than that in the initial ersatz urine and gradually increased from 177.3 mg L^−1^ to a value of 223.4 mg L^−1^ with the feed temperature over the tested range (Figure 3a), demonstrating that a higher proportion of phenol and water vapor across the membrane than that in the feed stream (i.e., *β* > 1). The *β* increased with increasing temperature over the range of 57 °C to 69 °C. Similar results were also found regarding the concentration of phenol by MD [36]. Compared with the temperature, no obvious change in the phenol concentration was found with the increasing permeate side pressure (Figure 3b), suggesting that the separation factor negligibly changed with the increasing vacuum degree over the tested range of pressures.

The electric conductivity as the combined effect of various trace quantities of electrolytes (including ammonia, phenol, and hydrogen chloride) was maintained at 16.28–19.32 μs cm^−1^, and no obvious change was found with the feed temperature and permeate side pressure. Considering the slight influence of phenol on the electricity of aqueous solutions (Section 3.4), ammonium and chloride ions were thus the main components of the electrolytes in the distillate (because of the detected TAN and low pH, the latter was attributable to the volatile hydrogen chloride dissolved in the distillate water). 

### 3.4. Influence of Volatile Components on the Distillate Water Quality

Although VMD has an almost 100% theoretical rejection for nonvolatile solutes, this rejection cannot prevent the permeation of volatile substances into distillate water [24]. The major volatile components in stabilized urine include free ammonia, volatile organics (e.g., phenol, formic acid, and acetic acid), and free hydrogen chloride [21]. The three components are volatile as free molecules and yet nonvolatile as ions. Ammonia has two forms in urine, free ammonia and ammonium ions, the proportion of which depends on the pH (Appendix A). The increasing pH in the aqueous solution favored the formation of free ammonia. Therefore, the TAN concentration in the distillate water substantially increased with the increasing volatility of the free ammonia in the feedstock when the pH increased from 5.5–9.5 (Figure 4a).

The conductivity of the distillate water was dominated by various trace quantities of electrolytes (including ammonia, phenol, and hydrogen chloride). However, the tested conductivity exhibited a positive correlation with the TAN concentration over the pH range of 5.5–9.5 in the feedstock, suggesting that the conductivity in the distillate was mainly dominated by the ammonium ions deriving from the volatile free ammonia from the feedstock. The resulting OH^−^ by the ionization of ammonia (Appendix A) also resulted in the slightly alkaline nature of the distillate when the pH of the feedstock was 5.5–9.5. Moreover, the distillate water was acidic when the pH of the feed was <3.5, suggesting that some nonionized hydrogen chloride volatilized through the micropores and was ionized in the distillate, although free ammonia was prevented in the rather acidic conditions.

The TOC concentration of the distillate water was close to that of the carbon content of phenol (Figure 4a), indicating that phenol was the major carbon source in the distillate. To further clarify the influence of phenol on the TOC concentration, the conductivity, and the pH of the distillate water, a comparative study was performed by VMD of ersatz urine without phenol (the other components were identical to the ersatz urine in Appendix A). The TOC concentration was <5 mg L^−1^ in the distillate (Figure 4b), which further confirmed that the phenol constituted the major organic carbon species in the distillate water. Moreover, the hypothesis that a low concentration of phenol had little impact on the pH and conductivity due to the low ionization degree was tested by the measured pH and conductivity of the phenol standard solution (Appendix A).

### 3.5. Influence of Water Recovery Rate on the Distillate Flux and Quality

The permeate flux substantially increased with the operation time in the initial stage (ca. 10 h, corresponding to a water recovery rate of ca. 32% in Figure 5), which resulted from minor changes in the membrane morphology [37,38]. Thereafter, the permeate flux slightly decreased from 3.70 L m^2^ h^−1^ to a value of 3.45 L m^−2^ h^−1^ until the water recovery rate reached ca. 85.3%. This result was attributable to the decreased vapor pressure due to the concentration of the feed [39]. Thereafter, a substantial decrease in the permeate flux was found, suggesting that the deposition and aggregation of foulants onto the membrane surface covered the membrane pores. Moreover, the initial permeate flux of the M2 module was 3.24–3.70 L m^−2^ h^−1^, which was ca. 35% of that using the M1 module at the same feed temperature (feed average pressure, T_m_) and permeate side pressure. This phenomenon resulted from the temperature polarization [40,41,42]. During the VMD of the ersatz urine, heat was required for the phase transition at the evaporation surface—which caused heat transfer from the feed flux to the evaporation surface in the membrane pores and formed a temperature gradient from the bulk feed stream to the evaporation surface in the micropores (Appendix A). In accordance with convective heat transfer theory, the heat transfer coefficient exhibited a positive relationship with the flow rate of the feed stream [40,41]. Because the flow rate in the M2 membrane module was only a tenth of that in the M1 module, the smaller heat transfer coefficient in the M2 module corresponded to a greater difference in temperature between the bulk feed stream and the evaporation surface. Therefore, less permeate flux was produced due to a lower saturated vapor pressure at the same temperature as the bulk feed stream, in accordance with Antoine’s equation (Appendix A). Although the permeate rate substantially decreased with an increasing module scale, there was a slight influence of the module dimensions due to the high surface area to volume ratio of the hollow fiber membrane. 

The TAN concentration was <5 mg L^−1^ throughout the VMD due to the negligible diffusion of volatile ammonia through the membrane, which is consistent with the aforementioned tests. The TOC concentration gradually decreased from 219.3 mg L^−1^ to a value of 58.4 mg L^−1^ with the increasing water recovery rate, which resulted from the proportion of phenol to water in the permeate flux exceeding that in the feed stream (i.e., *β* > 1 in Equation (5)) in the initial stage. With an increasing water recovery rate, the phenol concentration in the permeate stream gradually decreased due to the decreasing content of phenol in the remained feed stream. The pH gradually deceased from 5.92 to a value of 3.98, and the electric conductivity increased from 14.04 μs cm^−1^ to a value of 37.17 μs cm^−1^ within the same period. These results are attributable to the hydrogen chloride dissolved in the distillate water. In accordance with the concentrating feed steam, more hydrogen chloride permeated through the membrane pores and dissolved into H^+^ and Cl^−^, which resulted in decreasing pH and increasing conductivity.

To confirm the speculation of the fouling deposition on the membrane and to investigate the influence of fouling on the distillate water quality, the samples collected from the module after 91% water recovery were detected by SEM and EDS. The SEM images demonstrate that the fouled layer deposited on the inner surface of the membrane, which formed amorphous scaling and crystals that covered the feed surface (Appendix A). The scaling layer resulted in a gradual decrease in the permeate flux by decreasing the area of the active membrane surface when the water recovery exceeded 85.3%. Compared with the inner surface (feed side), there was negligible fouling on the outer surface (permeate side) of any of the membrane samples (Appendix A), suggesting that no nonvolatile solutes permeated in the downstream and led to the deterioration of the distillate water quality. The result indicated that the diffusion rate variation of the volatile components across the membrane, rather than the nonvolatile solute leakage, caused the changes of TOC, conductivity, and pH in the distillate water in accordance with the increasing water recovery rate. Elemental analysis of the scaling layer indicates that the membrane experienced substantial organic and inorganic fouling in accordance with the feedstock concentration (Appendix A). Fouling—especially scaling—accelerates the performance degradation of VMD [40,43,44]. Thus, the water recovery rate was limited to 80% in the multi-batch VMD tests over the prolonged service time of the membrane modules. 

### 3.6. Operation Stability of Urine VMD

Figure 6 shows the change in the permeate flux with the increasing number of batches. Substantial fluctuation of the permeate flux in different batches was observed, suggesting some fouling formed at cooling and in the salt-concentrated conditions during the shutdown for reloading [45]. 

However, the permeate flux did not consistently decrease and was nearly recovered by replacing the feedstock in the initial stage, suggesting that the fouled layer was removed due to cleaning by the replenished feed stream. The conductivity of the distillate water was at a relatively low level during the initial 10 batches; thereafter, the conductivity moderately increased. Thus, the developing membrane wetting resulted in a trace quantity of nonvolatile electrolyte leakage. Furthermore, the TOC concentration in the distillate water (the TOC mainly originated from the dissolved phenol in the distillate water, the average concentration in each batch of produced water) stabilized at 135–144 mg L^−1^ in the initial stage. The TOC increased to 240 mg L^−1^ after 30 batches, suggesting trace quantities of nonvolatile organic transfer with the feed leakage. The TAN concentration remained <5 mg L^−1^ over the initial 18 batches and slightly increased to 14.3 mg L^−1^ by batch 30. Thus, a small quantity of ammonium ions transferred into the distillate water. In summary, the removal efficiency of the TOC and TAN reached 94.2% and 95.8%, respectively. However, the increase in the conductivity of the distillate water, from 16.06 μs to a value of 125 μs cm^−1^, indicates the substantially degraded character of the multi-batch operation of urine VMD. 

When the VMD operation exceeded 22 batches, a discernible decrease in the permeate flux was observed. Meanwhile, some black-grey deposits—a visual sign of wetting—were evident on the outer surface of the membrane (Appendix A). The SEM images indicated that the foulants on the outer surface of the membrane formed amorphously layered scaling with a thickness of 5–25 μm (Figure 7a,b and Figure 8).

Compared with the virgin membrane, there were no obvious changes in terms of the morphology and dimensions of the pores on the inner surface of the used membrane sample (Figure 7d and Appendix A), suggesting that the microstructure was not destroyed after 30 batches of VMD. The contact angle at the stained position (Appendix A) decreased from ca. 105° to a value of 86° (ca. 18% reduction compared with the virgin membrane), which indicates that the aqueous ions and organics easily transferred with the leaking liquid water—and compromised the product water quality—due to the deteriorated hydrophobicity. An interesting phenomenon was the scaling deposition on the outer surface (permeate side) rather than the inner surface (feed side) (Figure 7c,d), which indicates that little insoluble fouling formed at the feed surface when the water recovery rate was limited to 80%. One explanation for the scaling on the permeate side is that the aqueous ions and organics transferred through the membrane with the leaking feed due to the wetting of some of the pores; thereafter, the nonvolatile solutes precipitated—concomitant with the water vaporization on the permeate surface. Although the deterioration of the water quality resulted from feed leakage, the moderate decrease in the permeate flux indicates that only a small fraction of the pores were wetted rather than a large extent of leakage after 30 batches of urine VMD, because excessive leakage would have resulted in an increasing flux [46,47]. 

Elemental analysis indicated that the major elements in the deposited layer were S, O, C, N, Cl, and some metal elements (Figure 8). To attain a better understanding of the fouling on the membrane, the functional groups in the virgin and used membrane samples (on the scaling position) were characterized by FTIR spectroscopy. The typical symmetric and nonsymmetric stretching vibration absorptions of –CH_3_ and –CH_2_ were observed at the 2867, 2838, 2971, and 2950 cm^−1^, respectively (Figure 9) [48]. The peaks at 1455 and 1376 cm^−^^1^ corresponded to the nonsymmetric angular vibrations of –CH_3_ and the symmetric angular vibrations of –CH_2_ [48]. The aforementioned results demonstrate the typical infrared spectrum that is characteristic of atactic polypropylene [48]. For the used PP membranes, some new peaks in the spectrum gave functional group information on the scaling layer. The broad peak at 3213 cm^−1^ was attributable to the coupling effect of the stretching vibration of –NH_2_ in the amide III adsorption band, with a stretching vibration of –OH in the carboxyl [49]. The adsorption peaks at 1616 cm^−^^1^ were attributable to the bending vibration of –C=O in the amide II adsorption band. The broad peak at 1450 cm^−^^1^ was attributed to the stretching vibration of –NH_2_ in the amide I adsorption, the bending vibration of –OH in the carboxyl, and the bending vibration of –CH_3_ [49]. The adsorption peak at 1238 cm^−^^1^ was attributable to –CH_3_ [50]. Thus, urea and some organic acid deposited in the fouled layer. In addition, the strong peak at 1100 cm^−^^1^, as well as the peaks at ca. 1270 and 615 cm^−^^1^, correspond to the SO_4_^2−^ group [51]. The FTIR analysis further confirmed that salt and some organic matter transferred through the wetting membrane pores, which resulted in the increased TOC and TAN concentrations as well as the increased conductivity of the distillate. 

No foulant deposition into the membrane wall matrix was observed by morphological and elemental linear analysis (Figure 8), which indicates that fouling on the wet pores and the performance degradation were inconspicuous after 30 batches of VMD tests [47]. In other words, the gradual pore wetting was mainly responsible for the deterioration of the permeate flux and the water quality. According to previous studies, the increasing wall thickness could delay the wetting process, which would counteract the feed leakage and enable longer operation and thus produce high-quality distillate water. Moreover, membrane cleaning (e.g., appropriate cleaner and frequency) results in a greater hydrophobic recovery for the surface fouling than that in the membrane wall. These measures can delay the wetting process and reduce the need to switch to a new module [52,53]. 

## 4. Conclusions

Regarding this study’s tests of water recovery from stabilized urine by VMD, the pH of the feedstock was the dominant factor for the distillate water quality. A low pH reduced the electrical conductivity and the TAN in the distillate. However, volatile components (e.g., phenol) could not be removed by VMD, which constituted the remaining organic carbon in the produced water.

With the increasing water recovery rate, the conductivity of the distillate slightly increased with the concentration of hydrogen chloride in the feedstock. When the water recovery rate exceeded 85.3%, a substantial decrease in the permeate flux resulted from membrane scaling on the feed side surface.

A gradual decline in the water quality was observed after 10 batches of urine VMD. Stepwise membrane wetting followed by minor feed leakage was responsible for the phenomenon. Nevertheless, the distillate water met the requirements of, e.g., oxygen formation by electrolysis and flushing water in the ISS. Moreover, VMD can serve as a primary treatment module—embedded in a wastewater treatment system, in a similar way to a VCD module. Chromium was not detected in the distillate water. However, the further removal (e.g., ion-exchange) and precise measure of the cation are required if the distillate is to be used as drinking water, due to the toxicity of hexavalent chromium; this is just like the way it is handled on the ISS, unless it is replaced by another low toxic bacteriostat. Considering the light weight and high reliability of the membrane module, VMD is a promising and feasible alternative for water reclamation from urine. 

## Figures and Tables

**Figure 1 membranes-12-00629-f001:**
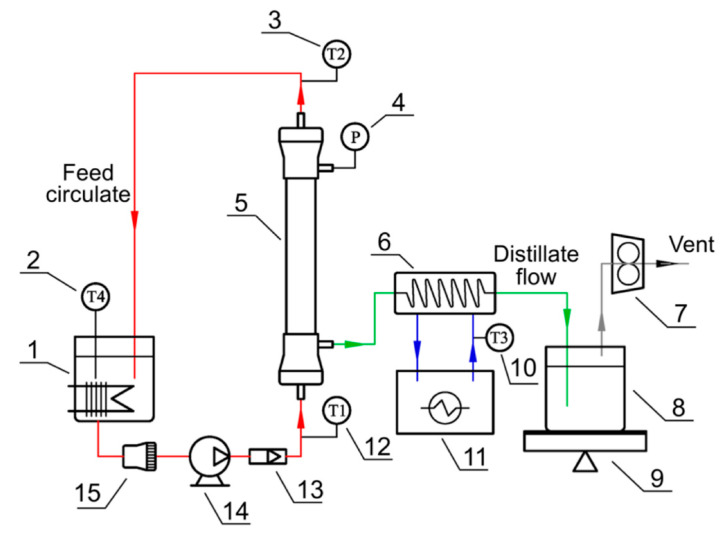
Schematic of the vacuum membrane distillation (VMD) system for stabilized urine treatment: 1, feed tank with heater; 2, 3, 10, 12, thermometers (T1 and T2: membrane module inlet and outlet; T3: cooling water; T4: feedstock); 4, vacuum transducer; 5, membrane module; 6, coil condenser; 7, vacuum pump; 8, distillate tank (sealed); 9, online balance; 11, cryostat; 13, water flow meter; 14, magnetic pump; 15, melt-blown filter (25 μm).

**Figure 2 membranes-12-00629-f002:**
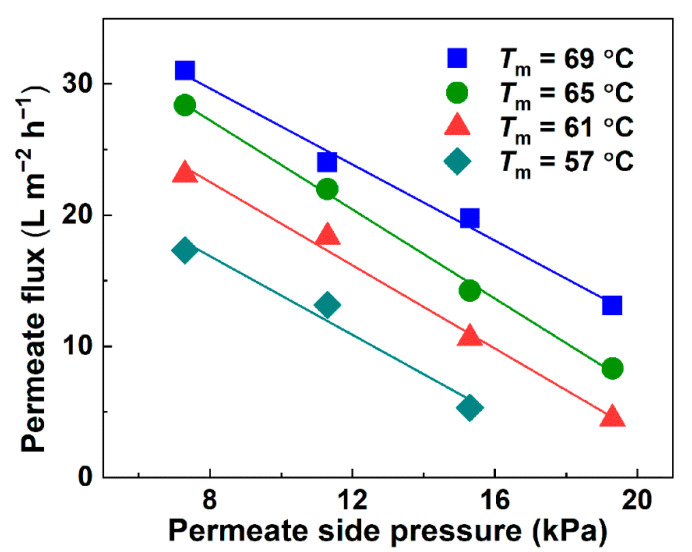
Permeate flux of ersatz urine VMD at various temperatures and permeate side pressures.

**Figure 3 membranes-12-00629-f003:**
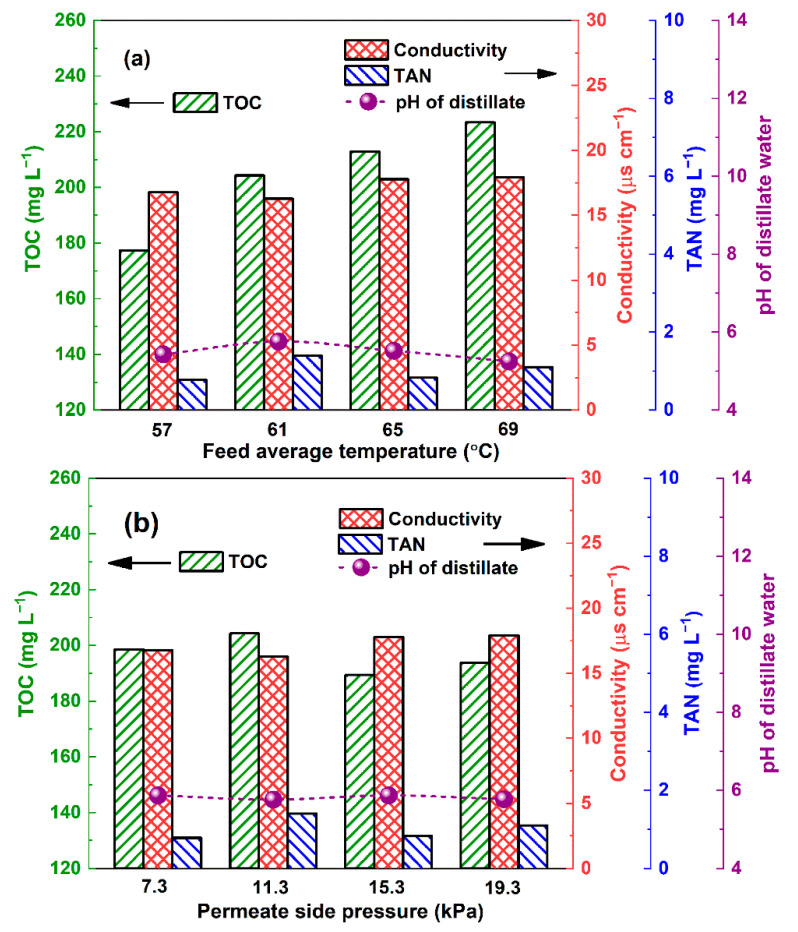
Change in TOC and TAN concentrations as well as conductivity and pH in distillate water with feed average temperature (**a**) and permeate side pressure (**b**) (permeate side pressure for former test: 11.3 kPa; feed average temperature for latter test: 61 °C, initial pH of feed: 1.6).

**Figure 4 membranes-12-00629-f004:**
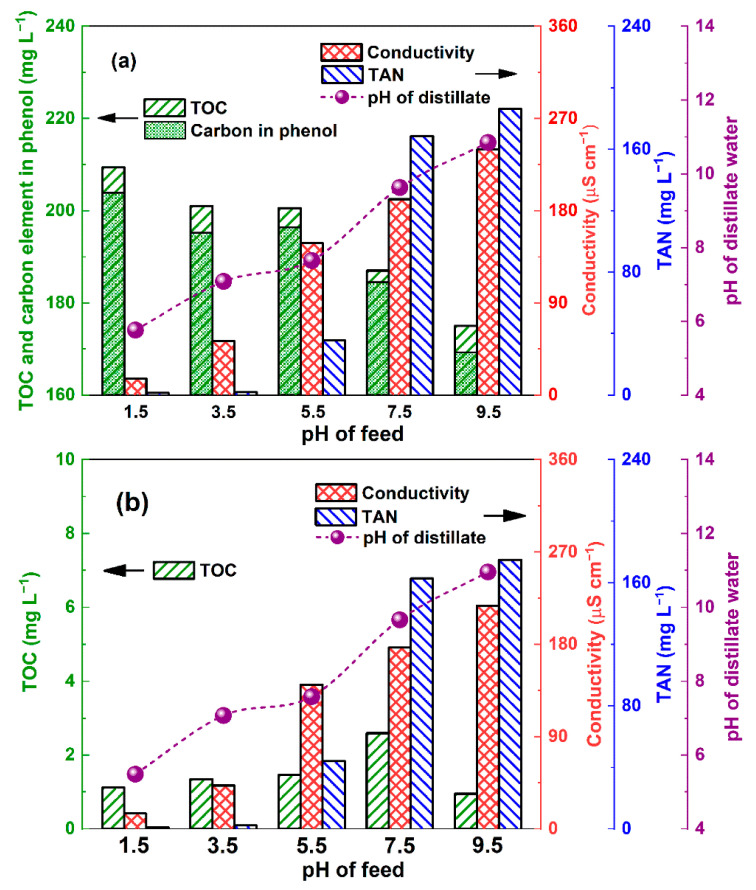
Change in TOC and TAN concentrations as well as conductivity and pH in distillate water with initial pH of feed with phenol (**a**) and without phenol (**b**) (feed average temperature: 61 °C, permeate side pressure: 11.3 kPa).

**Figure 5 membranes-12-00629-f005:**
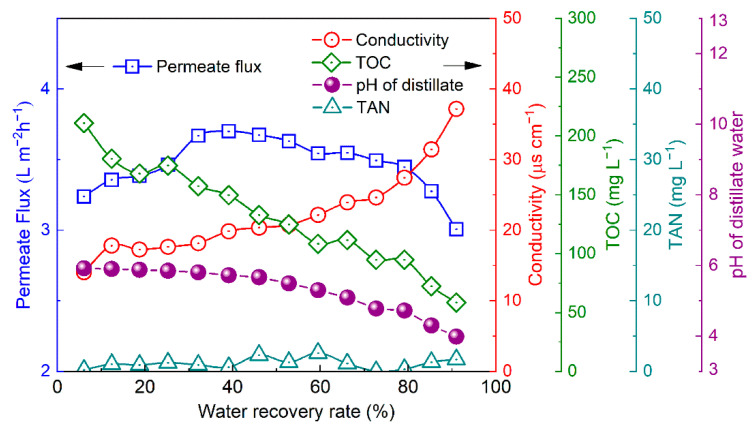
Changes in permeate flux, conductivity, and pH, as well as the concentrations of TOC and TAN with the water recovery rate (initial feed pH: 1.6, average temperature: 61 °C, permeate side pressure: 15.3 kPa).

**Figure 6 membranes-12-00629-f006:**
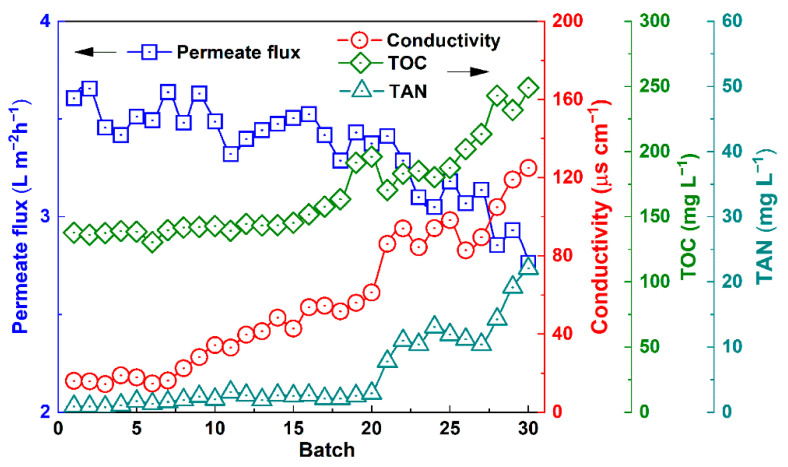
Changes in the distillate flux and conductivity, as well as the concentrations of TOC and TAN during the 30-batch operation (feed pH: 1.6, feed average temperature: 61 °C, permeate side pressure: 15.3 kPa).

**Figure 7 membranes-12-00629-f007:**
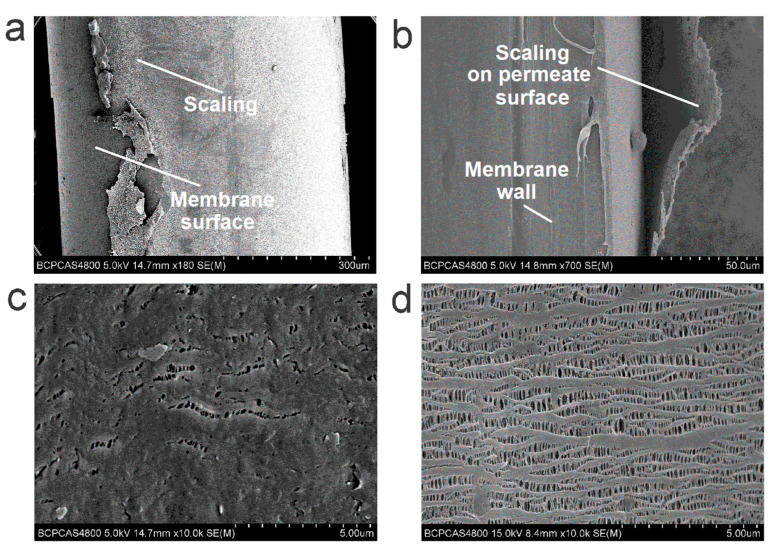
Morphological analysis of membrane samples after 30 batches of VMD tests. Representative morphology of the fouled layer on the outer surface (**a**) and cross-section (**b**); SEM images of the membrane on the outer surface (**c**) and inner surface (**d**).

**Figure 8 membranes-12-00629-f008:**
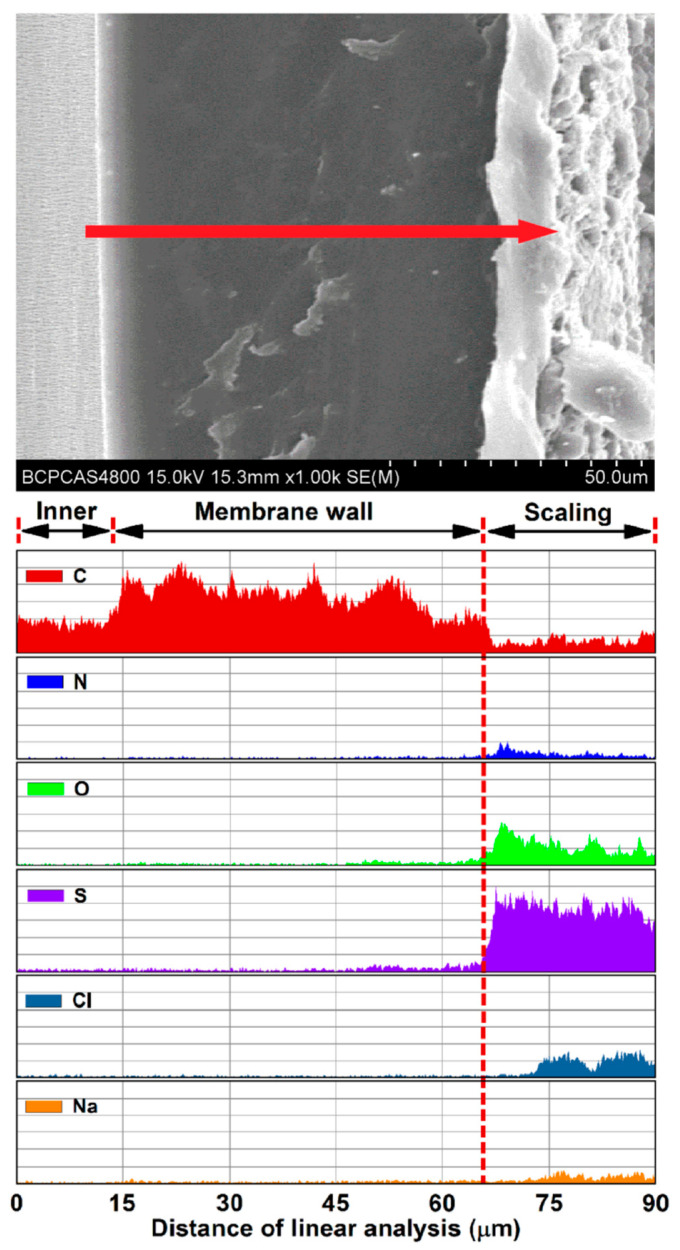
Elemental linear analysis of a cross-section of the fouled membrane sample.

**Figure 9 membranes-12-00629-f009:**
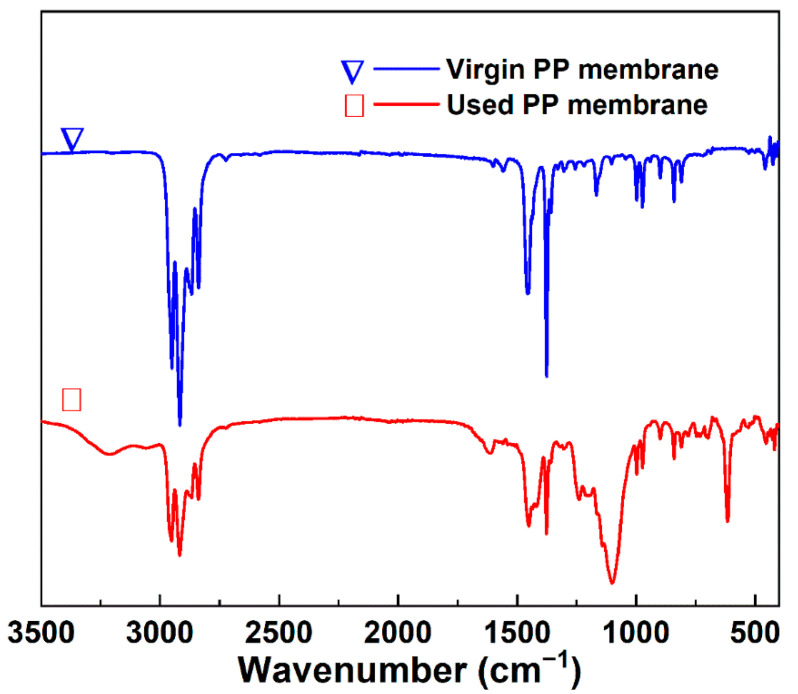
FTIR spectra of the virgin (blue line marked with a triangle) and used (red line marked with a square) PP hollow fiber membrane samples.

**Table 1 membranes-12-00629-t001:** Data on fresh and stabilized ersatz urine ^a^.

Index ^b^	Ersatz Urine	Stabilized Ersatz Urine
pH	6.0–7.0	1.5–2.0
Conductivity	21 ± 1.0	32 ± 1.0
Plate count (CFU/mL)	2.2 × 10^5^	—
TOC (mg/L)	4271.5 ± 53.31	4247 ± 19.20
TAN (mg/L)	340 ± 12.73	337 ± 15.35
Urea (mg/L)	14,376.6 ± 576.36	14,321.79 ± 321.31
TP (mg/L)	341.4 ± 15.34	268.75 ± 35.55
Phenol (mg/L)	164.32 ± 11.15	159.44 ± 10.93
Na^+^ (mg/L)	3227.91 ± 108.22	3194.15 ± 106.36
K^+^ (mg/L)	1569.91 ± 27.89	1580.65 ± 118.56
Mg^2+^ (mg/L)	125.03 ± 3.72	121.99 ± 7.25
Ca^2+^ (mg/L)	14.39 ± 0.26	4.145 ± 0.80
Cr (mg/L)	—	202.5 ± 6.73
SO_4_^2^^−^ (mg/L)	673.255 ± 14.14	5130.94 ± 112.27
Cl^−^ (mg/L)	6325.69 ± 164.47	6344.12 ± 315.22

^a^ Ersatz urine and stabilized ersatz urine after adding 17% flush water. ^b^ CFU, colony-forming units; TAN, total ammonia nitrogen; TOC, total organic carbon; TP, total phosphorus; Cr, chromium element.

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
