# Peer review of "High-Efficiency Water Recovery from Urine by Vacuum Membrane Distillation for Space Applications: Water Quality Improvement and Operation Stability"

_membranes, 2022, doi:10.3390/membranes12060629_

Round 1
Reviewer 1 Report
Please, check attached doc for reviewer's comments.

Author Response
Responds to reviewer comments
Title: High-Efficiency Water Recovery from Urine by Vacuum Membrane Distillation for Space Applications: Water Quality Improvement and Operation stability
Manuscript: membranes-1728901
Authors: Fei Wang1,2, Junfeng Liu1, Da Li1, Zheng Liu3, Jie Zhang1, Ping Ding2, Guochang Liu3 and Yujie Feng1*
Membranes
Revision due before: 2-Jun-2022.
Point 1: The submitted paper addresses a very specific problem. This can limit the paper audience but, on the other hand, it serves as an example of the application of membrane distillation technology, which is not widely employed industrially. The English writing needs to be revised. The design of experiments is enough to achieve the paper goals. I recommend the publication of this article after a revision of minor points.
Response1: Thanks for your comment. Before the first submission, we have also commissioned a professional language service (LetPub) for the manuscript polishing. The certificate provided by LetPub can be found as attached file. According to the suggestion, we re-checked the manuscript carefully to find any improper expression in the manuscript and revised many sites mainly in section Abstract, Introduction and Materials and Methods. Given that the revised sites discretely distributed here and there, they are not listed in the response letter.
Point2:.Page 2, line 92: Please, I kindly recommend the authors to discuss/justify better the addition of phenol to the urine to simulate a volatile organic component. Why it is needed to study the effects of this kind of component?
Response2: Thanks for your suggestion. According to the previous studies, the membrane distillation (MD) applied a microporous hydrophobic membrane for separation of the feed solution and downstream produced by the condensate of permeated water vapor. Almost complete rejection of nonvolatile solutes in the feed was achieved by MD process, whereas which could be ineffective to reject the volatile components. Urine is a complex wastewater containing some volatile components including phenol, formic acid and acetic acid etc. [1,2](below Reference), among which phenol exhibits the maximum concentration in urine according to the previous study. When the MD was used as the water recovery from urine, the volatile components would obviously influence on the water quality. So, to investigate the effects of volatile components on MD process of urine, the phenol was added to ersatz urine as the representative volatile organic component in this study.
To clarify the reason of the addition of phenol to the urine, we added some description in the revised manuscript as follows:
Page 2 section 2.2:
To investigate the effects of volatile components on the distillate quality during urine VMD, the phenol exhibiting the maximum concentration than other volatile constituents, was added as a representative in ersatz urine [21].
Reference
- Bouatra, S.; Aziat, F.; Mandal, R.; An, C.G.; Wilson, M.R.; Knox, C.; Bjorndahl, T.C.; Krishnamurthy, R.; Saleem, F.; Liu, P.; etc. The human urine metabolome. PLoS ONE 2018, 8, e73076- e73076.
- Putnam, D. Composition and Concentrative Proprieties of Human Urine; NASA: Washington, DC, 1971.
Point3: Page 6, section 3.2: It is discussed the influence of some operation parameters on permeate flux. However, the experiment used in the studies are dynamic, and it is expected any oscillation in the flux during time. Then, I kindly suggest authors to present how the permeate flux presented in Figure 2 was calculated. Is it an average?
Response3: Thanks for your comment. According to our study, the permeate flux was oscillated and was affected by the water recovery rate due to the concentration of the feed. To eliminate these influences, firstly, the experiments at each condition (feed temperature and permeate side pressure) were performed at three times and the average was used as analysis and description. Secondly, when the influence of the operating conditions (e.g. feed temperature, pH, and permeate side pressure) on the VMD was studied, the water recovery rate was limited to <30% to eliminate the influence of the concentration of the feed.
To clarify the consideration to reduce the influence of oscillation on the permeate flux, we added some description in Materials and Methods section in the revised manuscript as follows:
Page 4 section 2.5:
The tests of permeate flux were performed at three times and the average was used as discussion.
Point4: Page 11, line 376: Please, explain better the sentence “…suggesting that a lack of penetration of nonvolatile solutes deteriorated the water quality”.
Response4: Thanks for your comment. To clearly indicate the results, we have revised the description according to your suggestion.
Page 11 section 3.5:
Compared with the inner surface (feed side), there was negligible fouling on the outer surface (permeate side) of any of the membrane samples (Figure S4a), suggesting that no nonvolatile solutes permeated in downstream and led to the deterioration of the distillate water quality. The result indicated that the diffusion rate variation of the volatile components across the membrane, rather than nonvolatile solute leakage, caused the changes of TOC, conductivity, and pH in the distillate water in accordance with increasing water recovery rate.
Reviewer 2 Report
The authors of the manuscript consider a relevant, but on the other hand, a specific topic regarding water purification in extreme conditions, for example, at the International Space Station. It is proposed to extract water from human waste. For this, it is proposed to use hollow fiber membranes molded from polypropylene melts. For the polymer used, I recommend that the authors indicate the characteristics or provide appropriate references. It is also desirable to provide information on the hollow fiber spinning process. After all, this stage is the key in the formation of the structural features of future membranes. The theoretical part of the manuscript is done at a decent level. The authors comprehensively and at the same time briefly outline the essence of the problem and possible ways to solve it. Such a description is well suited for both specialists and unprepared readers. The practical part is supplemented by a fairly large amount of additional materials. That is, in the course of the work, the authors involved numerous research methods, including structural ones. The conclusions of the manuscript fully reflect the main results of the work. In my opinion, the manuscript may be considered for publication by the editor. Below are some minor questions and recommendations:
Line 21. The abbreviation VMD must be deciphered at the first mention.
Table S2 and Figure S1. According to micrographs, the average pore diameter is higher than 282 ± 19. I recommend that the authors pay attention to this. The question also arises why the authors use the term pores rather than crazes or defects.
Line 414. thickness values ​​should be checked - "10–25 micron". The photographs give the impression that said layer has a thinner thickness.
Figure 9. I recommend reducing the displayed area of ​​wavenumber to 3500 cm-1. This will enlarge the drawing and make the main part more informative. At the moment, many bands are poorly distinguishable.
Line 444 onwards. "2867.0, 2838.0, 2971.3, and 2949.8" - values ​​can be rounded.
